# DEEP K-NN FOR NOISY LABELS

## ABSTRACT

Modern machine learning models are often trained on examples with noisy labels that hurt performance and are hard to identify. In this paper, we provide an empirical study showing that a simple $k$-nearest neighbor-based filtering approach on the logit layer of a preliminary model can remove mislabeled training data and produce more accurate models than some recently proposed methods. We also provide new statistical guarantees into its efficacy.

## 1 INTRODUCTION

Machine learned models can only be as good as the data they were used to train on. With increasingly large modern datasets and automated and indirect labels like clicks, it is becoming ever more important to investigate and provide effective techniques to handle noisy labels.

We revisit the classical method of filtering out suspicious training examples using $k$-nearest neighbors ($k$-NN) (Wilson, 1972). Like Papernot & McDaniel (2018), we apply $k$-NN on the learned intermediate representation of a preliminary model, which adds robustness. In fact, the $k$-nearest neighbor approach has recently been receiving attention for its robustness properties (Wang et al., 2018; Reeve & Kaban, 2019) and as an auxiliary strategy for modern machine learning (Jiang et al., 2018).

The main contributions of this paper are:

- Experimentally showing that identifying mislabeled examples by $k$-NN executed on an intermediate layer of a preliminary deep model works well compared to state-of-art methods for handling noisy labels across noise levels, and is robust to the choice of $k$.
- Theoretically showing that $k$-NN's predictions will only identify a training example as clean if its label is the Bayes-optimal label. We also provide finite-sample analysis in terms of the margin and how spread out the corrupted labels are (Theorem 1), rates of convergence for the margin (Theorem 2) and rates under Tsybakov's noise condition (Theorem 3) with all rates matching minimax-optimal rates in the noiseless setting.

Our work shows that even though the preliminary neural network is trained with corrupted labels, it still yields intermediate representations that are useful for $k$-nearest neighbor filtering. Given labels which are in high disagreement, one can either automatically remove them and retrain on the remaining, or send to a human operator for further review. This strategy is also be useful in human-in-the-loop systems where one can warn the human annotator that a label is suspicious, and automatically propose new labels based on its nearest neighbors' labels.

In addition to strong empirical performance, deep $k$-NN filtering has a couple of advantages. Firstly, many methods require a clean set of samples whose labels can be trusted. Here we show that the $k$-NN based method is robust and does not require such a clean set of samples. Second, while $k$-NN does introduce the hyperparameter $k$, we will show that deep $k$-NN filtering is stable to the choice of $k$: such robustness to hyperparameters is highly desirable as optimal tuning for this problem is often not available in practice (i.e. when no clean validation set is available).

## 2 RELATED WORK

We review relevant prior work in training on noisy labels and related $k$-NN theory.

## 2.1 TRAINING WITH NOISY LABELS

Methods to handle label noise can be classified into two main strategies: (i) explicitly identify and remove the noisy examples, and (ii) indirectly handle the noise with robust training methods.

**Data Cleaning** This proposal fits into the broad family of *data cleaning* methods, in that our proposal detects and filters *dirty data* (see Chu et al. (2016) for a recent survey). The use of $k$-NN to "edit" training data has been popular since Wilson (1972) used it to throw away training examples that were not consistent with their $k = 3$ nearest neighbors. The idea of using a preliminary model to help identify mislabeled examples dates to at least Guyon et al. (1994), who proposed using the model to compute an information gain for each example and then suspecting ones with high gain. Other early work used a cross-validation set-up to train a classifier on part of the data, then use it to make a prediction on held-out training examples, and remove any examples if the prediction disagrees with the label; ensembles of models can also be used for the predictions. (Brodley & Freidl, 1999).

**Noise Corruption Estimation** For multi-class problems, a popular approach is to account for noisy labels by applying a confusion matrix after the model's softmax layer (Sukhbaatar et al., 2014). Such methods rely on a confusion matrix which is often unknown and must be estimated. Patrini et al. (2017) suggest deriving it from the softmax distribution of the model trained on noisy data while Goldberger & Ben-Reuven (2016); Jindal et al. (2016); Han et al. (2018) give alternatives. Accurate estimates are generally hard to attain when only untrusted data is available. Hendrycks et al. (2018) achieves more accurate estimates in the setting where some amount of known clean, trusted data is available. Xiao et al. (2015); Khetan et al. (2017); Vahdat (2017) use EM-type algorithms to estimate the clean label distribution.

**Noise-Robust Training** Natarajan et al. (2013) propose a method to make any surrogate loss function noise-robust given knowledge of the corruption rates. Ghosh et al. (2017) proves that losses like mean absolute error (MAE) are inherently robust under symmetric or uniform label noise while Zhang & Sabuncu (2018) shows that training with MAE results in poor convergence and accuracy. They propose a new loss function based on the negative Box-Cox transformation that trades off the noise-robustness of MAE with the training efficiency of cross-entropy. Lastly, the ramp, unhinged, and savage losses have been proposed and theoretically justified to be noise-robust for support vector machines (Brooks, 2011; Van Rooyen et al., 2015; Masnadi-Shirazi & Vasconcelos, 2009). Rolnick et al. (2017) empirically shows that deep learning models are robust to noise when there are enough correctly labeled examples and when the model capacity and training batch size are sufficiently large.

**Auxiliary Models** Veit et al. (2017) propose learning a label cleaning network on trusted data by predicting the differences between clean and noisy labels. Li et al. (2017) suggests training on a weighted average between noisy labels and distilled predictions of an auxiliary model trained on trusted data.

**Example Weighting** Here we make a hard decision about whether to keep a training example, but one can also adapt the weights on training examples based on the confidence in their labels. Liu & Tao (2015) provides an importance-weighting scheme for binary classification. Ren et al. (2018) suggests upweighting examples whose loss gradient is aligned with those of trusted examples at every step in training. Jiang et al. (2017) investigates a recurrent network that learns a sample weighting scheme to give to the base model.

## 2.2 $k$-NEAREST NEIGHBOR THEORY

The theory of $k$-nearest neighbor classification has a long history, for example: Fix & Hodges Jr (1951); Cover (1968); Stone (1977); Devroye et al. (1994); Chaudhuri & Dasgupta (2014). Much of the prior work focuses on $k$-NN's statistical consistency properties. However, with the growing interest in adversarial examples and learning with noisy labels, there have recently been analyses of $k$-nearest neighbor methods in these settings. Wang et al. (2018) analyze the robustness of $k$-NN classification and provide a robust variant of 1-NN classification where their notion of robustness is that predictions of nearby points should be similar. Gao et al. (2016) provides an analysis of the $k$-NN classifier under noisy labels and like us, show that $k$-NN can attain similar rates in the noisy setting as in the noiseless setting. Gao et al. (2016) assumes a noise model where labels are

corrupted uniformly at random, while we assume an arbitrary corruption pattern and provide results based on a notion of how spread out the corrupted points are. Moreover, we provide finite-sample bounds borrowing recent advances in $k$-NN convergence theory in the noiseless setting (Jiang, 2019) while the guarantees of Gao et al. (2016) are asymptotic. Reeve & Kaban (2019) provide stronger guarantees on a robust modification of $k$-NN proposed by Gao et al. (2016). To the best of our knowledge, we provide the first finite-sample rates of consistency for the classical $k$-NN method in the noisy setting with very little assumptions on the label noise.

## 3 ALGORITHM

We first define the $k$-nearest neighbor classifier:

**Definition 1** ($k$-NN). *Let the k-NN radius of $x \in \mathcal{X}$ be $r_k(x) := \inf\{r : |B(x, r) \cap X| \geq k\}$ where $B(x, r) := \{x' \in \mathcal{X} : |x - x'| \leq r\}$ and the k-NN set of $x \in \mathcal{X}$ be $N_k(x) := B(x, r_k(x)) \cap X$. Then for all $x \in \mathcal{X}$, the k-NN classifier function w.r.t. $X$ has discriminant function*

$$\eta_k(y; x) := \frac{1}{|N_k(x)|} \sum_{i=1}^{n} 1 \left[y_i = y, \ x_i \in N_k(x)\right],$$

*with prediction $\eta_k(x) := \arg\max_y \eta_k(y; x)$.*

Our method Algorithm 1 assumes a dataset $\mathcal{D}_{\text{noisy}}$ with potentially noisy labels, along with a dataset $\mathcal{D}_{\text{clean}}$ consisting of clean or trusted labels. Note that we allow $\mathcal{D}_{\text{clean}}$ to be empty (i.e. in instances where no such trusted data is available). We have found that having $\mathcal{D}_{\text{clean}}$ becomes important when $\mathcal{D}_{\text{noisy}}$ has a high corruption rate; otherwise the representations learned by training on $\mathcal{D}_{\text{noisy}}$ alone are often reasonable enough. The procedure begins by training on either $\mathcal{D}_{\text{noisy}} \cup \mathcal{D}_{\text{clean}}$ or $\mathcal{D}_{\text{clean}}$. For our experiments, we partition $\mathcal{D}_{\text{clean}}$ into a training set $\mathcal{D}_{ct}$ and validation set $\mathcal{D}_{cv}$ and train models on $\mathcal{D}_{ct}$ and $\mathcal{D}_{\text{noisy}} \cup \mathcal{D}_{ct}$ and choose the one that performs better on $\mathcal{D}_{cv}$.

We then filter examples that disagree with the $k$-NN classifier prediction, where the $k$-NN is computed on the final logit layer of the trained model (i.e. the layer right before softmax).

---

**Algorithm 1** Filtering datapoints via deep $k$-NN.

---

**Inputs:** $\mathcal{D}_{\text{noisy}}, \mathcal{D}_{\text{clean}}, k$
Train model $\mathcal{M}$ on either $\mathcal{D}_{\text{noisy}} \cup \mathcal{D}_{\text{clean}}$ or $\mathcal{D}_{\text{clean}}$.
Let $\mathcal{N}$ be the activations of $\mathcal{D}_{\text{noisy}} \cup \mathcal{D}_{\text{clean}}$ on the logit layer of $\mathcal{M}$.
$\mathcal{D}_{\text{filtered}} := \{(x, y) \in \mathcal{N} : \eta_k(x) = y\}$, where $\eta_k$ is computed w.r.t. $\mathcal{N}$.
Train final model on $\mathcal{D}_{\text{filtered}} \cup \mathcal{D}_c$.

---

## 4 THEORETICAL ANALYSIS

For the theoretical analysis, we assume the binary classification problem with the features defined on compact set $\mathcal{X} \subseteq \mathbb{R}^D$. We assume that points are drawn according to distribution $\mathcal{F}$ as follows: the features come from distribution $\mathbb{P}_{\mathcal{X}}$ on $\mathcal{X}$ and the labels are distributed according to the measurable conditional probability function $\eta : \mathcal{X} \to [0, 1]$. That is, a sample $(X, Y)$ is drawn from $\mathcal{F}$ as follows: $X$ is drawn according to $\mathbb{P}_{\mathcal{X}}$ and $Y$ is chosen according to $\mathbb{P}(Y = 1|X = x) = \eta(x)$.

The goal will be to show that given corrupted examples, the $k$-NN disagreement method is still able to identify the examples whose labels do not match that of the Bayes-optimal label.

We will make a few regularity assumptions for our analysis to hold. The first regularity assumption ensures that the support $\mathcal{X}$ does not become arbitrarily thin anywhere. This is a standard non-parametric assumption (e.g. Singh et al. (2009); Jiang (2019)).

**Assumption 1** (Support Regularity). *There exists $\omega > 0$ and $r_0 > 0$ such that $Vol(\mathcal{X} \cap B(x, r)) \geq \omega \cdot Vol(B(x, r))$ for all $x \in \mathcal{X}$ and $0 < r < r_0$, where $B(x, r) := \{x' \in \mathcal{X} : |x - x'| \leq r\}$.*

Let $p_{\mathcal{X}}$ be the density function corresponding to $\mathbb{P}_{\mathcal{X}}$. The next assumption ensures that with a sufficiently large sample, we will obtain a good covering of the input space.

**Assumption 2** ($p_{\mathcal{X}}$ bounded from below). $p_{X,0} := \inf_{x \in \mathcal{X}} p_X(x) > 0.$

Finally, we make a smoothness assumption on $\eta$, as done in other analyses of $k$-NN classification (e.g. Chaudhuri & Dasgupta (2014); Reeve & Kaban (2019))

**Assumption 3** ($\eta$ Hölder continuous). *There exists $0 < \alpha \leq 1$ and $C_\alpha > 0$ such that $|\eta(x) - \eta(x')| \leq C_\alpha |x - x'|^\alpha$ for all $x, x' \in \mathcal{X}$.*

We propose a notion of how spread out a set of points is based on the minimum pairwise distance between the points. This will be a quantity in the finite-sample bounds we will present. Intuitively, the more spread out a contaminated set of points is, the less clean samples we will be needed to overcome the contamination of that set.

**Definition 2** (Minimum pairwise distance).

$$S_2(C) := \min_{x,x' \in C, x \neq x'} |x - x'|.$$

Also define the $\Delta$-interior region of $\mathcal{X}$ where there is at least $\Delta$ margin in the probabilistic label:

**Definition 3.** *Let $\Delta \geq 0$. Define $\mathcal{X}^\Delta := \{x \in \mathcal{X} : \left|\frac{1}{2} - \eta(x)\right| \geq \Delta\}.$*

We now state the result, which says that with high probability *uniformly* on $\mathcal{X}^\Delta$ when $\Delta > 0$ is known, we have that the label disagrees with the $k$-NN classifier if and only if the label is not the Bayes-optimal prediction. Due to space, all of the proofs have been deferred to the Appendix.

**Theorem 1** (Fixed $\Delta$). *Let $\Delta, \delta > 0$ and suppose Assumptions 1, 2, and 3 hold. There exists constants $K_l, K_u > 0$ depending only on $\mathcal{F}$ such that the following holds with probability at least $1 - \delta$. Let $X_{[n]}$ be $n$ (uncorrupted) examples drawn from the $\mathcal{F}$ and $C$ be a set of points with corrupted labels and denote our sample $X := X_{[n]} \cup C$. Suppose $k$ lies in the following range*

$$K_l \cdot \frac{1}{\Delta^2} \cdot \log^2(1/\delta) \cdot \log n \leq k \leq K_u \cdot \min\{S_2(C)^D, \Delta^{D/\alpha}\} \cdot n,$$

*then the following holds uniformly over $x \in \mathcal{X}^\Delta$: the k-NN prediction computed w.r.t. $X$ agrees with the label if and only if the label is the Bayes-optimal label $\eta^*(x) := 1[\eta(x) \geq \frac{1}{2}]$.*

In the last result, we assumed that $\Delta$ was fixed. We next show how we can make a similar guarantee but show that we can take $\Delta \to 0$ as we choose $k, n \to \infty$ appropriately and provide rates of convergence.

**Theorem 2** (Rates of convergence for $\Delta$). *Let $\delta > 0$ and suppose Assumptions 1, 2, and 3 hold. There exist constants $K_l, K_u, K > 0$ depending only on $\mathcal{F}$ such that the following holds with probability at least $1 - \delta$. Let $X_{[n]}$ be $n$ (uncorrupted) examples drawn from $\mathcal{F}$, and $C$ be a set of points with corrupted labels and denote our sample $X := X_{[n]} \cup C$. Suppose $k$ lies in the following range*

$$K_l \cdot \log^2(1/\delta) \cdot n^{\frac{\alpha}{\alpha+D}} \leq k \leq K_u \cdot S_2(C)^D \cdot n,$$

*then the following holds uniformly over $x \in \mathcal{X}^\Delta$: the k-NN prediction computed w.r.t. $X$ agrees with the label if and only if the label is the Bayes-optimal label $\eta^*(x) := 1[\eta(x) \geq \frac{1}{2}]$ where*

$$\Delta = K \cdot \left( \sqrt{\frac{\log n + \log(1/\delta)}{k}} + \left(\frac{k}{n}\right)^{\alpha/D} \right).$$

**Remark 1.** *Choosing $k = O(n^{2\alpha/(2\alpha+D)})$ in the above result gives us $\Delta = \widetilde{O}(n^{-\alpha/(2\alpha+D)})$. This rate for $\Delta$ is the minimax-optimal rate for $k$-nearest neighbor classification on $\mathcal{X}^\Delta$ given a sample of size $n$ (Chaudhuri & Dasgupta, 2014) in the uncorrupted setting. Thus, our analysis is tight up to logarithmic factors.*

We next give results with an additional margin assumption, also known as Tsybakov's noise condition (Mammen et al., 1999; Tsybakov et al., 2004):

**Assumption 4** (Tsybakov Noise Condition). *The following holds for some $C_\beta$ and $\beta$ and all $\Delta > 0$:*

$$\mathbb{P}_{\mathcal{X}}(x \notin \mathcal{X}^\Delta) \leq C_\beta \cdot \Delta^\beta.$$

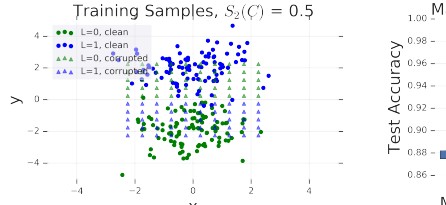 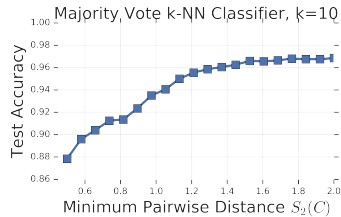 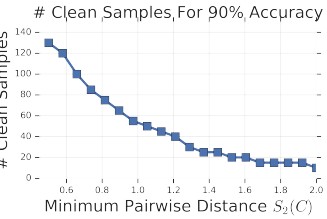

Figure 1: Left: training samples. We observe that test accuracy improves as $S_2(C)$ increases (middle) and that fewer clean training samples are needed to achieve an accuracy of 90% (right).

**Theorem 3** (Rates under Tsybakov Noise Condition). *Let $\delta > 0$ and suppose Assumptions 1, 2, 3 and 4 hold. There exists constants $K_l, K_u, K, K' > 0$ depending only on $\mathcal{F}$ such that the following holds with probability at least $1 - \delta$. Let $X_{[n]}$ be $n$ (uncorrupted) examples drawn from the $\mathcal{F}$ and $C$ be a set of points with corrupted labels and denote our sample $X := X_{[n]} \cup C$. Suppose $k$ lies in the following range*

$$K_l \cdot \log^2(1/\delta) \cdot n^{\frac{\alpha}{\alpha+D}} \le k \le K_u \cdot S_2(C)^D \cdot n,$$

*and define $\eta_k(x) := \arg\max_y \eta_k(y; x)$. Then,*

$$\mathbb{P}\left(\eta_k(x) \ne \eta^*(x)\right) \le K \cdot \left(\sqrt{\frac{\log n + \log(1/\delta)}{k}} + \left(\frac{k}{n}\right)^{\alpha/D}\right)^{\beta},$$

$$R_X - R^* \le K' \cdot \left(\sqrt{\frac{\log n + \log(1/\delta)}{k}} + \left(\frac{k}{n}\right)^{\alpha/D}\right)^{\beta+1}$$

*where $R_X := \mathbb{E}_{\mathcal{F}}[g_k(x) \ne y]$ and $R^* := \mathbb{E}_{\mathcal{F}}[g^*(x) \ne y]$ denote the risk of the $k$-NN method and Bayes optimal classifier, respectively.*

**Remark 2.** *Choosing $k = O(n^{2\alpha/(2\alpha+D)})$ in the above gives us a rate of $\widetilde{O}(n^{-\alpha(\beta+1)/(2\alpha+D)})$ for the excess risk. This matches the lower bounds of Audibert et al. (2007) up to logarithmic factors.*

### 4.1 IMPACT OF MINIMUM PAIRWISE DISTANCE

The minimum pairwise distance across corrupted samples, $S_2(C)$, is a key quantity in the theory presented in the previous section. We now empirically study its significance in a simulated binary classification task in 2 dimensions. Clean samples with label $L$ are generated by sampling i.i.d from $\mathcal{N}(\mu_L, I_{2 \times 2})$, where $\mu_0 = (0, -2)$ and $\mu_1 = (0, 2)$. The decision boundary is the line $y = 0$. We take 100 samples uniformly spaced on a square grid centered about $(0, 0)$ and corrupt them by flipping their true label. With this construction, $S_2(C)$ is precisely the grid width, which we let vary. The training set is a union of 100 clean samples and the 100 corrupted samples. Using 1000 clean samples as a test set we study the classification performance of a majority vote $k$-NN classifier, where $k = 10$. Results are shown in Figure 1. As expected, we see that as $S_2(C)$ decreases, so does test accuracy and we need more clean training samples to compensate.

## 5 EXPERIMENTS

We evaluate the effectiveness of our algorithm as follows. We split each dataset's training set into two parts, $\mathcal{D}_{\text{clean}}$ and $\mathcal{D}_{\text{noisy}}$. We then corrupt the labels of some fraction of examples in $\mathcal{D}_{\text{noisy}}$ by applying a corruption matrix prescribed by one of the following methods.

- **Uniform**: The label is flipped to any one of the labels (including itself) with equal probability.
- **Flip**: The label is flipped to any *other* label with equal probability.
- **Hard Flip**: With probability $\frac{1}{2}$, we flip the label $m$ to $\pi(m)$ where $\pi$ is some predefined permutation of the labels.

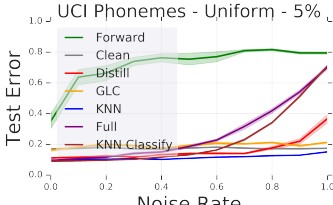 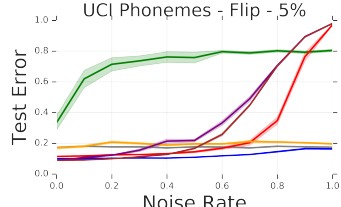 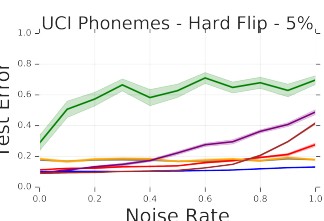

Figure 2: **UCI Results**. Error plots against amount of noise applied to the labels of $\mathcal{D}_{\text{noisy}}$. $\mathcal{D}_{\text{clean}}$ contains $5\%$ of the data. Each column is a different corruption and each row is for a different dataset. We see that the $k$-NN method consistently chooses the best datapoints to filter leading to lower error. More results are in the Appendix.

We compare against the following baselines:

- **Gold Loss Correction (GLC)** (Hendrycks et al., 2018) estimates the corruption matrix by averaging the softmax outputs of the clean examples on a model trained on noisy data.
- **Distill** (Li et al., 2017) assigns each example in the combined dataset a "soft" label that is a convex combination of its label and its softmax output from a model trained solely on clean data.
- **Forward** (Patrini et al., 2017), similar in spirit to GLC, estimates the corruption matrix by training a model on noisy data and using the softmax output for prototype examples for each class. It does not require a clean dataset like other methods.
- **Clean.** We define this as training on the clean data only.
- **Full.** We define this as training on the full (clean and noisy) data.
- $k$-**NN Classify** is like "Full" except we use $k$-NN majority voting on the logits layer for classification at test time.

We report test errors and show the average across multiple runs with standard error bands shaded. Errors are computed on 11 uniformly distributed noise rates between 0 and 1 inclusive. For the results shown in the main text, we have that $\mathcal{D}_{\text{clean}}$ is randomly selected and is $5\%$ of the data. In the Appendix, we show results over different sizes of $\mathcal{D}_{\text{clean}}$. We implement all methods using the Tensorflow 2.0 Keras API and Scikit-Learn. We use the Adam optimizer with default learning rate 0.001 and a batch size of 128 across all experiments. For the UCI datasets, we set $k = 50$ and set $k = 500$ for all other datasets. We chose $k = 50$ for the UCI datasets because some of the datasets were of small size. However, we found that the $k$-NN method's performance was quite stable to the choice of $k$, which we show in Section 5.4. We describe the permutations used for hard flipping in the Appendix.

### 5.1 UCI AND MNIST RESULTS

We show the results for one of the UCI datasets in Figure 2 and Fashion MNIST in Figure 3. Due to space, results for MNIST and the remaining UCI datasets are in the Appendix. For UCI, we use a fully-connected neural network with a single hidden layer of dimension 100 with ReLU activations and train for 100 epochs. For both MNIST datasets, we use is a two hidden-layer fully-connected neural network where each layer has 256 hidden units with ReLU activations. We train the model for 20 epochs. We see that the $k$-NN approach attains models with a low error rate across noise rates and either outperforms or is competitive with the next best method, GLC.

### 5.2 CIFAR RESULTS

For CIFAR10/100 we use ResNet-20, which we train from scratch on single NVIDIA P100 GPUs. We train CIFAR10 for 100 epochs and CIFAR100 for 150 epochs. We show results for CIFAR10 in Figure 4 and results for CIFAR100 in the Appendix, due to space. We see that the $k$-NN method performs competitively. It generally outperforms on the uniform and flip noise types but performs

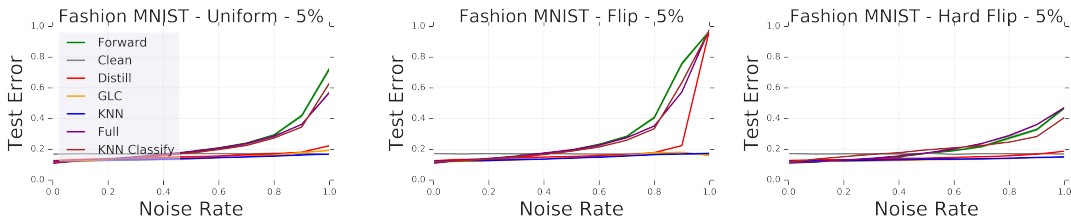

Figure 3: **Fashion MNIST**. Each column is a different corruption method. We see that the $k$-NN approach performs competitively. More results are in the Appendix.

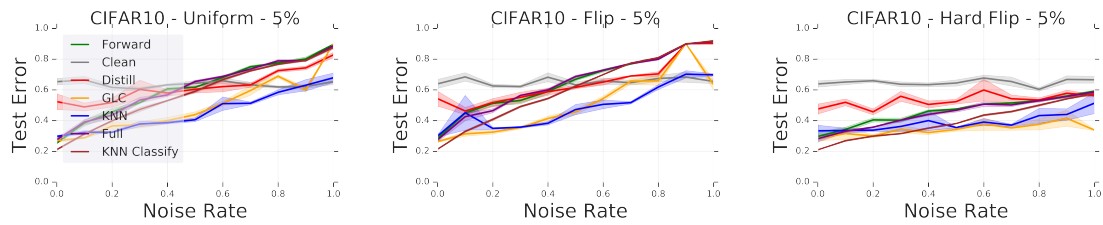

Figure 4: **CIFAR10**. Each column is a different corruption method. We see that our $k$-NN method performs competitively or outperforms on the uniform and flip noise types but performs worse for the hard flip noise type. More results are in the Appendix.

worse for the hard flip noise type. It is not too surprising that $k$-NN would be weaker in the presence of hard flip noise (i.e. where labels are mapped based on a pre-determined mapping between labels) as the noise is much more structured in that case making it more difficult to be filtered out by majority vote among the neighbors. In other words, unlike the uniform and flip noise types, we are no longer dealing with *white label noise* in the hard flip noise type.

## 5.3 SVHN RESULTS

We show the results in Figure 5. We train ResNet-20 from scratch on NVIDIA P100 GPUs for 100 epochs. As in the CIFAR experiments, we see that the $k$-NN method tends to be competitive in the uniform and flip noise types but does slightly worse in the hard flip.

## 5.4 ROBUSTNESS TO $k$

In this section, we show that our procedure is stable in its hyperparameter $k$. The theoretical results suggest that a wide range of $k$ can give us statistical consistency guarantees and we show that in practice a wide range of $k$ gives us similar results for Algorithm 1 (Figure 6). Such robustness in

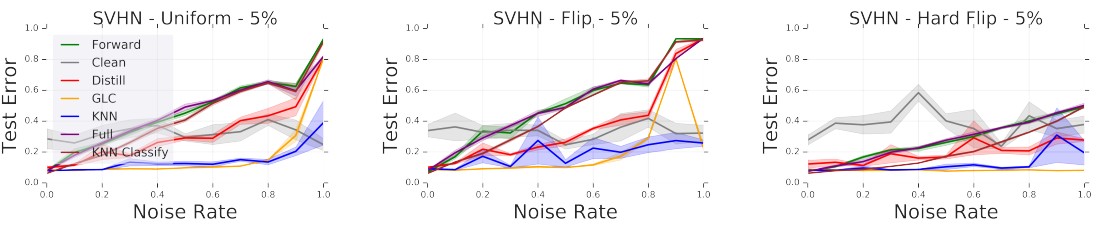

Figure 5: **SVHN**. We see that the $k$-NN method performs competitively on the uniform and flip noise types but performs worse for the hard flip noise type. More results in the Appendix.

| Dataset | % Clean | Uniform | | | | | | | Flip | | | | | | | Hard Flip | | | | | | |
|---|---|---|---|---|---|---|---|---|---|---|---|---|---|---|---|---|---|---|---|---|---|---|
| | | Forward | Clean | Distill | GLC | k-NN | k-NN Classify | Full | Forward | Clean | Distill | GLC | k-NN | k-NN Classify | Full | Forward | Clean | Distill | GLC | k-NN | k-NN Classify | Full |
| Letters | 5 | 4.55 | 3.16 | 2.48 | 2.33 | **2.05** | 2.19 | 2.61 | 4.83 | 3.16 | 2.96 | 2.35 | **2.1** | 2.52 | 2.92 | 3.77 | 3.17 | 2.1 | 1.83 | **1.82** | 1.82 | 2.52 |
| | 10 | 4.28 | 2.51 | 2.05 | 1.91 | **1.78** | 1.79 | 2.23 | 4.45 | 2.52 | 2.12 | 1.92 | **1.8** | 2.06 | 2.49 | 3.63 | 2.52 | 1.85 | 1.6 | **1.6** | 1.64 | 2.36 |
| | 20 | 3.77 | 2.06 | 1.76 | 1.57 | 1.56 | **1.34** | 1.85 | 3.85 | 2.07 | 1.78 | 1.56 | 1.58 | **1.42** | 1.93 | 3.35 | 2.05 | 1.62 | **1.38** | 1.41 | 1.42 | 2.04 |
| Phonemes | 5 | 7.89 | 1.91 | 1.79 | 2.12 | **1.26** | 2.58 | 3 | 7.91 | 1.93 | 3.21 | 2.16 | **1.34** | 3.97 | 4.31 | 6.6 | 1.92 | 1.77 | 1.96 | **1.2** | 1.78 | 2.7 |
| | 10 | 7.86 | 1.54 | 1.53 | 1.67 | **1.16** | 2.28 | 2.85 | 7.95 | 1.54 | 2.75 | 1.69 | **1.22** | 3.64 | 3.96 | 6.73 | 1.55 | 1.61 | 1.6 | **1.15** | 1.62 | 2.58 |
| | 20 | 7.72 | 1.34 | 1.33 | 1.35 | **1.13** | 1.76 | 2.24 | 7.89 | 1.34 | 1.97 | 1.35 | **1.16** | 2.87 | 3.28 | 6.36 | 1.33 | 1.45 | 1.3 | **1.14** | 1.38 | 2.28 |
| Wilt | 5 | 5.18 | 0.56 | 0.85 | 0.54 | **0.39** | 0.93 | 1.86 | 5.27 | 0.56 | 3.89 | 0.53 | **0.52** | 5.15 | 4.95 | 4.6 | 0.55 | 0.73 | 0.58 | **0.39** | 0.98 | 1.9 |
| | 10 | 4.68 | 0.43 | 0.75 | 0.45 | **0.32** | 0.77 | 1.77 | 5.63 | 0.44 | 3.14 | 0.43 | **0.41** | 4.86 | 4.84 | 4.78 | 0.43 | 0.66 | 0.43 | **0.31** | 0.78 | 1.81 |
| | 20 | 4.31 | 0.36 | 0.86 | 0.35 | **0.32** | 0.57 | 1.5 | 5.18 | 0.34 | 2.67 | 0.35 | **0.34** | 4.23 | 4.32 | 5.63 | 0.34 | 0.61 | 0.36 | **0.3** | 0.57 | 1.49 |
| Seeds | 5 | 3.29 | 4.22 | 3.71 | 4.2 | 3.08 | 2.87 | **2.71** | 5.13 | 4.33 | 5.56 | 4.39 | **3.64** | 5.11 | 4.88 | 2.94 | 4.06 | 3.63 | 3.91 | 2.99 | **2.84** | 3.01 |
| | 10 | 3.43 | 3.04 | 2.84 | 2.99 | **2.14** | 2.74 | 2.57 | 5.02 | 3.38 | 4.58 | 3.19 | **2.65** | 4.9 | 4.73 | 2.75 | 3.42 | 2.96 | 3.14 | **2.25** | 2.69 | 2.86 |
| | 20 | 2.99 | 2.69 | 2.27 | 2.74 | **1.72** | 2.44 | 2.2 | 4.41 | 2.56 | 3.57 | 2.65 | **2.09** | 4.23 | 3.86 | 2.85 | 2.53 | 2.25 | 2.57 | **1.62** | 2.43 | 2.49 |
| Iris | 5 | 2.97 | 3.23 | 3.48 | 3.97 | 2.46 | **1.72** | 2.05 | 5.25 | 3.38 | 5.15 | 4.03 | **3.02** | 4.32 | 4.43 | 2.9 | 3.29 | 3.13 | 4.1 | 2.32 | **1.14** | 1.35 |
| | 10 | 2.89 | 2.48 | 2.59 | 2.25 | 1.32 | **1.26** | 1.55 | 5.06 | 2.53 | 4.58 | 2.24 | **2.17** | 3.98 | 4.08 | 2.6 | 2.34 | 2.02 | 1.86 | 1.15 | **0.91** | 1.16 |
| | 20 | 2.46 | 1.85 | 1.97 | 1.52 | **0.6** | 0.96 | 1.32 | 4.46 | 1.51 | 3.45 | **1.38** | 1.46 | 3.36 | 3.53 | 2.51 | 1.84 | 1.48 | 1.34 | **0.58** | 0.65 | 1 |
| Parkinsons | 5 | 5 | 3.46 | 3.26 | 4.22 | 3.4 | **3.26** | 3.76 | 5.17 | 3.55 | 4.49 | 4.1 | **3.36** | 5.34 | 5.35 | 4.98 | 3.71 | 3.56 | 4.59 | 3.68 | **3.28** | 3.63 |
| | 10 | 5.35 | 3.26 | 3.22 | 3.45 | **3.21** | 3.22 | 3.82 | 5.43 | 3.38 | 4.25 | 3.44 | **3.32** | 5.27 | 5.13 | 5.24 | 3.26 | 3.1 | 3.37 | **3.1** | 3.12 | 3.82 |
| | 20 | 4.88 | 3.01 | 3.08 | 3.1 | 2.98 | **2.97** | 3.52 | 5.19 | 3.02 | 3.94 | 3.05 | **2.95** | 4.99 | 5.06 | 5.1 | 2.98 | 3.01 | 2.98 | 2.96 | **2.91** | 3.47 |
| MNIST | 5 | 2.88 | 0.69 | 1.03 | 0.5 | **0.4** | 2.72 | 2.75 | 3.6 | 0.69 | 1.91 | 0.5 | **0.44** | 3.46 | 3.49 | 2.03 | 0.69 | 0.78 | **0.22** | 0.29 | 0.65 | 2.14 |
| | 10 | 2.57 | 0.5 | 0.85 | 0.41 | **0.33** | 2.42 | 2.45 | 3.22 | 0.5 | 1.5 | 0.42 | **0.35** | 3.1 | 3.14 | 1.86 | 0.5 | 0.67 | **0.21** | 0.26 | 0.48 | 1.98 |
| | 20 | 2.07 | 0.35 | 0.69 | 0.34 | **0.27** | 1.97 | 2.03 | 2.48 | 0.35 | 0.86 | 0.34 | **0.27** | 2.36 | 2.41 | 1.54 | 0.35 | 0.53 | **0.2** | 0.22 | 0.36 | 1.67 |
| Fashion MNIST | 5 | 2.76 | 1.88 | 1.73 | 1.59 | **1.56** | 2.53 | 2.54 | 3.55 | 1.87 | 2.55 | **1.59** | 1.6 | 3.3 | 3.31 | 2.3 | 1.87 | 1.62 | **1.44** | 1.48 | 2.31 | 2.4 |
| | 10 | 2.47 | 1.71 | 1.6 | 1.52 | **1.52** | 2.21 | 2.3 | 3.14 | 1.71 | 2.13 | **1.53** | 1.54 | 2.92 | 2.99 | 2.14 | 1.71 | 1.52 | **1.41** | 1.46 | 2.19 | 2.22 |
| | 20 | 2.07 | 1.56 | 1.48 | 1.45 | **1.44** | 1.95 | 2.05 | 2.38 | 1.56 | 1.54 | **1.46** | 1.46 | 2.17 | 2.27 | 1.92 | 1.56 | 1.43 | **1.38** | 1.41 | 2.04 | 1.97 |
| CIFAR10 | 5 | 6.74 | 7 | 6.86 | 5.43 | **5.03** | 6.34 | 6.74 | 7.14 | 7.2 | 7.13 | 5.52 | **5.35** | 6.71 | 7.12 | 5.08 | 7.13 | 5.85 | **3.76** | 4.27 | 4.33 | 4.96 |
| | 10 | 6.58 | 6.58 | 6.32 | 5.39 | **5.27** | 6.11 | 6.55 | 6.82 | 6.56 | 6.72 | 5.62 | **5.32** | 6.48 | 6.83 | 4.89 | 6.53 | 5.3 | **3.91** | 4.4 | 4.21 | 4.89 |
| | 20 | 6.4 | 5.52 | 5.66 | 5.11 | **4.57** | 5.93 | 6.36 | 6.62 | 6.62 | 5.59 | 5.9 | 5.16 | **4.85** | 6.1 | 4.77 | 5.45 | 4.68 | **3.51** | 3.82 | 4.03 | 4.75 |
| CIFAR100 | 5 | 10.8 | 10.22 | 9.98 | 9.59 | 9.57 | **9.17** | 9.29 | 10.79 | 10.24 | 10.03 | 9.64 | 9.66 | **9.2** | 9.29 | 10.64 | 10.23 | 9.88 | 8.58 | 8.98 | **7.48** | 8.04 |
| | 10 | 10.79 | 9.94 | 9.7 | 9.42 | 9.63 | **9.09** | 9.25 | 10.81 | 9.89 | 9.68 | 9.46 | 9.63 | **9.1** | 9.25 | 10.65 | 9.89 | 9.38 | 8.56 | 9.19 | **7.44** | 7.99 |
| | 20 | 10.78 | 9.38 | 9.15 | 9.06 | 9.23 | **8.92** | 9.07 | 10.8 | 9.44 | 9.13 | 8.97 | 9.17 | **8.92** | 9.09 | 10.66 | 9.41 | 8.65 | 8.07 | 8.81 | **7.33** | 7.9 |
| SVHN | 5 | 5.04 | 3.52 | 3.56 | 1.99 | **1.62** | 4.64 | 4.95 | 5.6 | 3.65 | 4.09 | 2.17 | **2.05** | 5.23 | 5.52 | 3.04 | 4.17 | 2.16 | **0.89** | 1.35 | 2.32 | 3.04 |
| | 10 | 4.98 | 2.2 | 3.2 | 2.27 | **1.34** | 4.51 | 4.82 | 5.5 | 2.28 | 3.72 | 2.29 | **1.48** | 4.96 | 5.29 | 2.98 | 2.41 | 2.01 | **0.88** | 1.04 | 2.16 | 2.98 |
| | 20 | 4.32 | 1.83 | 2.67 | 2.14 | **1.2** | 3.96 | 4.41 | 4.85 | 1.83 | 3.09 | 1.9 | **1.19** | 4.34 | 4.82 | 2.7 | 1.85 | 1.54 | **0.98** | 0.98 | 1.83 | 2.74 |

Table 1: **Area under the test error vs noise rate curve**. Each row corresponds to a dataset and size of clean dataset $\mathcal{D}_{\text{clean}}$ pair, where the size is a percentage of the total training set (5%, 10%, 20%). Each column shows the area under the error curve across noise rates for a particular method and noise type (Uniform, Flip, Hard Flip). We see that the $k$-NN method consistently outperforms the other methods for Uniform and Flip and outperforms the other methods on Hard Flip on the smaller datasets.

hyperparameter is highly desirable because optimal tuning is often not available, especially when no sufficient clean validation set is available.

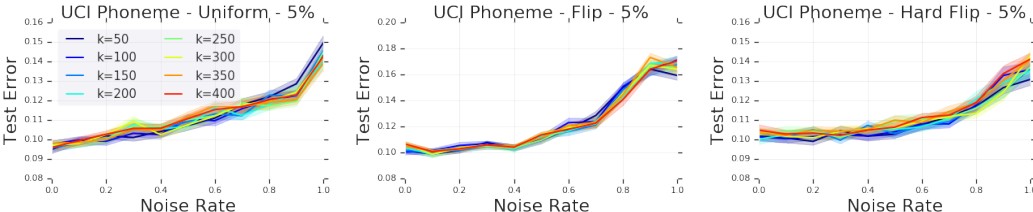

Figure 6: **Performance across different values of** $k$. Here we show that on a UCI dataset, the performance of Algorithm 1 is stable when varying its hyperparameter $k$. Note that the $y$-axis has been zoomed in to better see the differences between the curves.

## 5.5 AREA UNDER ERROR VS NOISE LEVEL CURVE ACROSS DATASETS

In the figures shown so far, it may be difficult to compare the curves in some cases so we report an area under the curve metric in Table 1.

**Conclusions and Open Questions** We conclude from our experiments and theory that the $k$-NN based method (Algorithm 1) is a relatively safe method to remove problematic training examples before training. While $k$-NN methods can be sensitive to the choice of $k$ when used with small datasets (Garcia et al., 2009), we hypothesize that with today's large datasets one can blithely set $k$ to a fixed practically medium-sized value (e.g. $k = 500$) as done here and expect reasonable performance. Theoretically we provided some new results for how well $k$-NN can identify clean versus corrupted labels. Open theoretical questions are whether there are alternate notions of how to characterize the difficulty of a particular configuration of corrupted examples and whether we can provide both upper and lower learning bounds under these noise conditions.

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

# A    PROOFS

## A.1    SUPPORTING THEORETICAL RESULTS

The following bounds $r_k(x)$ uniformly in $x \in \mathcal{X}$.

**Lemma 1** (Lemma 2 of Jiang (2019)). *The following holds with probability at least $1 - \delta/2$. If*

$$2^8 \cdot D \log^2(4/\delta) \cdot \log n \le k \le \frac{1}{2} \cdot \omega \cdot p_{X,0} \cdot v_D \cdot r_0^D \cdot n,$$

*then $\sup_{x \in \mathcal{X}} r_k(x) \le \left( \frac{2k}{\omega \cdot v_D \cdot n \cdot p_{X,0}} \right)^{1/D}$, where $v_D$ is the volume of the unit ball in $\mathbb{R}^D$.*

The next result bounds the number of distinct $k$-NN sets over $\mathcal{X}$.

**Lemma 2** (Lemma 3 of Jiang (2019)). *Let $M$ be the number of distinct $k$-NN sets over $\mathcal{X}$, that is, $M := |\{N_k(x) : x \in \mathcal{X}\}|$. Then $M \le D \cdot n^D$.*

## A.2    MINIMUM $k$-NN SPREAD

We propose a more notion of how spread out a set of points is than $S_2$ which will be used in the theoretical analysis. This will allow us to more precisely characterize how difficult a configuration of incorrectly labeled examples will be to work with in the $k$-NN context. For example, if such examples are spread out far apart, then there will be many correctly labeled examples nearby for the $k$-NN approach to identify the incorrectly labeled examples. On the other hand, if the corrupted examples are all close together, then it will be more difficult to identify them without many uncorrected examples in that region. To this end, we define the minimum $k$-NN spread:

**Definition 4** (minimum $k$-NN spread).

$$S_k(C) := \min_{x \in C} r_k(x, C),$$

*where $r_k(x, C)$ denotes the distance from $x$ to the $k$-th closest neighbor in $C$.*

Note that this definition is consistent with the earlier definition of $S_2$.

## A.3    PROOF OF THEOREM 1

*Proof of Theorem 1.* Let $\tau, \gamma, \epsilon > 0$ be quantities that will be determined later. Suppose that for some $x \in \mathcal{X}^\Delta$, we have $r_k(x) \le \tau$ and $S_{\lfloor (\frac{1}{2} - \gamma) \cdot k \rfloor}(C) \ge \tau$. Then, at least $\frac{1}{2} + \gamma$ fraction of the points within $x$'s $k$-nearest neighbors are not in the corrupted set $C$. Let $A_x := N_k(x) \backslash C$, that is, the $k$-nearest neighbors of $x$ that are not in $C$. Then it is clear that $A_x$ is a $k_0$-nearest neighbor set of $x$ relative to $X \backslash C$ for some $k_0 \ge \lceil (\frac{1}{2} + \gamma) \cdot k \rceil$. We have that $A_x \subseteq \mathcal{X}^\Delta \oplus \tau$ where $A \oplus r := \{x \in \mathcal{X} : \inf_{a \in A} |x - a| \le r\}$. Let us consider without loss of generality that $\eta(x) \ge \frac{1}{2} + \Delta$ (call this set $\mathcal{X}^{\Delta,+}$). The case $\mathcal{X}^{\Delta,-} := \{x \in \mathcal{X}^\Delta : \eta(x) \le \frac{1}{2} - \Delta\}$ follows by symmetry. Thus, we have $\eta(x') \ge \frac{1}{2} + \Delta - C_\alpha \tau^\alpha$ for all $x' \in A_x$. By Hoeffding's inequality, we have

$$\mathbb{P}\left( \frac{1}{|A_x|} \sum_{x' \in A_x} y(x') < \frac{1}{2} + \Delta - C_\alpha \tau^\alpha - \epsilon \right) \le \exp(-2\epsilon^2 \cdot k_0),$$

where $y(x)$ is the label corresponding to sample $x$. By Lemma 2, we have that there are at most $D \cdot n^D$ such $k_0$-nearest neighbor sets across all $k_0$ in $X \backslash C$. That is, this is also a bound on the number of distinct $A_x$ for $x \in \mathcal{X}$. Therefore, if we set

$$\epsilon = \sqrt{\frac{D \log n + \log(4D/\delta)}{(1 + 2\gamma) \cdot k}},$$

then by union bound, we have that

$$\mathbb{P}\left( \inf_{x \in \mathcal{X}^{\Delta,+}} \frac{1}{|A_x|} \sum_{x' \in A_x} y(x') < \frac{1}{2} + \Delta - C_\alpha \tau^\alpha - \epsilon \right) \le \frac{\delta}{4}.$$

and thus, with probability at least $1 - \delta/4$, we have that $\frac{1}{|A_x|} \sum_{x' \in A_x} y(x') \geq \frac{1}{2} + \Delta - C_\alpha \tau^\alpha - \epsilon$ *uniformly* over $x \in \mathcal{X}^{\Delta,+}$. Similarly, with probability at least $1 - \delta/4$ we have that $\frac{1}{|A_x|} \sum_{x' \in A_x} y(x') \leq \frac{1}{2} - \Delta + C_\alpha \tau^\alpha + \epsilon$ *uniformly* over $x \in \mathcal{X}^{\Delta,-}$.

Hence, in order for $k$-nearest neighbor prediction to predict the Bayes-optimal label on $\mathcal{X}^\Delta$, it suffices that

$$k_0 \left( \frac{1}{2} + \Delta - C_\alpha \tau^\alpha - \epsilon \right) \geq \frac{k}{2}.$$

Since $k_0 \geq (\frac{1}{2} + \gamma) \cdot k$, we have that the above holds if

$$\Delta \geq C_\alpha \tau^\alpha + \epsilon + \frac{1 - 2\gamma}{2(1 + 2\gamma)}.$$

We now choose the values of $\tau, \gamma, \epsilon$ to upper bound each of the terms on the R.H.S. by $\Delta/3$ so that the above holds.

We can bound the last term by $\Delta/3$ by setting:

$$\gamma = \frac{1}{2} \cdot \frac{3 - 2\Delta}{3 + 2\Delta}.$$

Next, taking

$$k \geq \frac{3(3 + 2\Delta)}{2\Delta^2} \left( D \log n + \log(4D/\delta) \right),$$

we have that $\epsilon \leq \Delta/3$. Now, by Lemma 1, we have that setting

$$\tau = \left( \frac{2k}{\omega \cdot v_D \cdot n \cdot p_{X,0}} \right)^{1/D}$$

gives us that $r_k(x) \leq \tau$ for all $x \in \mathcal{X}$ with probability at least $1 - \delta/2$. It thus suffices to take

$$k \leq \frac{1}{2} \left( \frac{\Delta}{3 \cdot C_\alpha} \right)^{D/\alpha} \cdot \omega \cdot v_D \cdot p_{X,0} \cdot n$$

so that $C_\alpha \tau^\alpha \leq \Delta/3$. Now in order for $S_{\lfloor (\frac{1}{2} - \gamma) \cdot k \rfloor}(C) \geq \tau$, it suffices to have $S_2(C) \geq \tau$. This can be accomplished by having the following hold:

$$k \leq \frac{1}{2} \cdot S_2(C)^D \cdot \omega \cdot v_D \cdot p_{X,0} \cdot n.$$

Thus, there exists positive constants $K_l$ and $K_u$ depending only on $\mathcal{F}$ such that if

$$K_u \cdot \frac{1}{\Delta^2} \cdot (\log^2(1/\delta) \cdot \log n) \leq k \leq K_u \cdot \min\{S_2(C)^D, \Delta^{D/\alpha}\} \cdot n,$$

then the desired conditions hold. $\qquad \square$

### A.4 Proof of Theorem 2

*Proof of Theorem 2.* The proof begins in the same way as the proof of Theorem 1. As before, let $\tau, \gamma, \epsilon > 0$ be quantities that will be determined later. Like before, we are reduced to showing

$$\Delta \geq C_\alpha \tau^\alpha + \epsilon + \frac{1 - 2\gamma}{2(1 + 2\gamma)},$$

as long as the conditions for Lemma 1 and 2 hold and $S_2(x) \geq \tau$ and $r_k(x) \leq \tau$ where we choose

$$\epsilon = \sqrt{\frac{D \log n + \log(4D/\delta)}{(1 + 2\gamma) \cdot k}}, \quad \gamma = \frac{1}{2} \cdot \frac{3 - 2\Delta}{3 + 2\Delta}, \quad \tau = \left( \frac{2k}{\omega \cdot v_D \cdot n \cdot p_{X,0}} \right)^{1/D}.$$

These conditions hold for some $K_u$ and $K_l$ depending on $\mathcal{F}$. Then we are reduced to having

$$\frac{2}{3}\Delta \geq \sqrt{\frac{D\log n + \log(4D/\delta)}{(1+2\gamma)\cdot k}} + C_\alpha\left(\frac{2k}{\omega\cdot v_D\cdot n\cdot p_{X,0}}\right)^{\alpha/D}.$$

Since $\gamma \geq 0$, it suffices to have

$$\frac{2}{3}\Delta \geq \sqrt{\frac{D\log n + \log(4D/\delta)}{k}} + C_\alpha\left(\frac{2k}{\omega\cdot v_D\cdot n\cdot p_{X,0}}\right)^{\alpha/D}.$$

The desired form for $\Delta$ clearly follows for some choice of $K$ depending only on $D, \omega, p_{X,0}, C_\alpha$, all of which depend only on $\mathcal{F}$.

Finally, we must ensure that $\lfloor(\frac{1}{2}-\gamma)\cdot k\rfloor \geq 2$ so that $S_{\lfloor(\frac{1}{2}-\gamma)\cdot k\rfloor}(C) \geq S_2(x)$. Given the expression for $\gamma$, it is equivalent to have $\lfloor\left(\frac{2\Delta}{3+2\Delta}\right)\cdot k\rfloor \geq 2$. It suffices to show that $k \geq \frac{3(3+2\Delta)}{2\Delta}$. Given the form of $\Delta$ in terms of $n$ and $k$, we see that it suffices to have that

$$k \geq \frac{9}{2}\cdot K\cdot\left(\left(\sqrt{\frac{\log n + \log(1/\delta)}{k}} + \left(\frac{k}{n}\right)^{\alpha/D}\right)\right)^{-1} + 3,$$

which holds when $k \geq K_0\cdot n^{2\alpha/(2\alpha+D)}$ for some $K_0$ depending only on $\mathcal{F}$, as desired. $\qquad\square$

### A.5 PROOF OF THEOREM 3

*Proof of Theorem 3.* The first part follows from Theorem 2. For the second part, we have by Theorem 2 that if $x \in \mathcal{X}^\Delta$, then the $k$-NN classifier and the Bayes-optimal classifier match with probability $1-\delta$ uniformly. Thus, we have

$$R_X - R^* \leq \mathbb{P}(x \notin \mathcal{X}^\Delta)\left(\mathbb{E}_\mathcal{F}[g_k(x) \neq y|x \notin \mathcal{X}^\Delta] - \mathbb{E}_\mathcal{F}[g^*(x) \neq y|x \notin \mathcal{X}^\Delta]\right)$$
$$\leq C_\beta\cdot\Delta^\beta\cdot\left(\mathbb{E}_\mathcal{F}[g_k(x) \neq y|x \notin \mathcal{X}^\Delta] - \mathbb{E}_\mathcal{F}[g^*(x) \neq y|x \notin \mathcal{X}^\Delta]\right) \leq C_\beta\cdot\Delta^\beta\cdot 2\Delta.$$

The result follows immediately from Theorem 2. $\qquad\square$

## B HARD FLIP PERMUTATIONS

For Fashion MNIST we hard flip by swapping the following classes: TSHIRT $\leftrightarrow$ SHIRT, TROUSER $\leftrightarrow$ DRESS, PULLOVER $\leftrightarrow$ COAT, SANDAL $\leftrightarrow$ BAG, SNEAKER $\leftrightarrow$ ANKLEBOOT. For CIFAR10 we swap the pairs: TRUCK $\leftrightarrow$ AUTOMOBILE, BIRD $\leftrightarrow$ AIRPLANE, DEER $\leftrightarrow$ HORSE, CAT $\leftrightarrow$ DOG, FROG $\leftrightarrow$ SHIP. For CIFAR100, we hard flip circularly (i.e. $\pi(i) = (i+1) \mod K$) within each of the 20 superclasses. For all other datasets, we hard flip circularly.

## C ADDITIONAL PLOTS

We provide the plots that were ommitted from the main text due to space constraints.

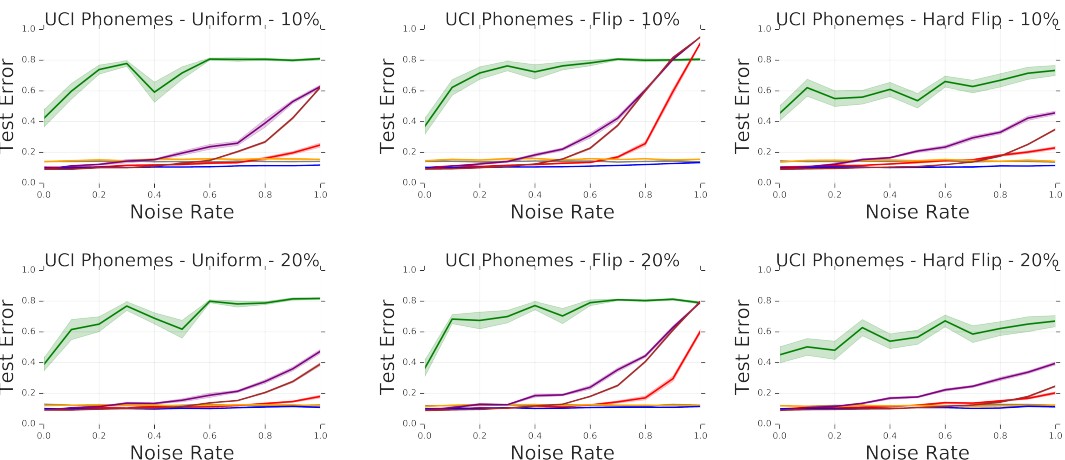

Figure 7: Plots for UCI Phonemes dataset at 10, 20% clean data and all corruption types.

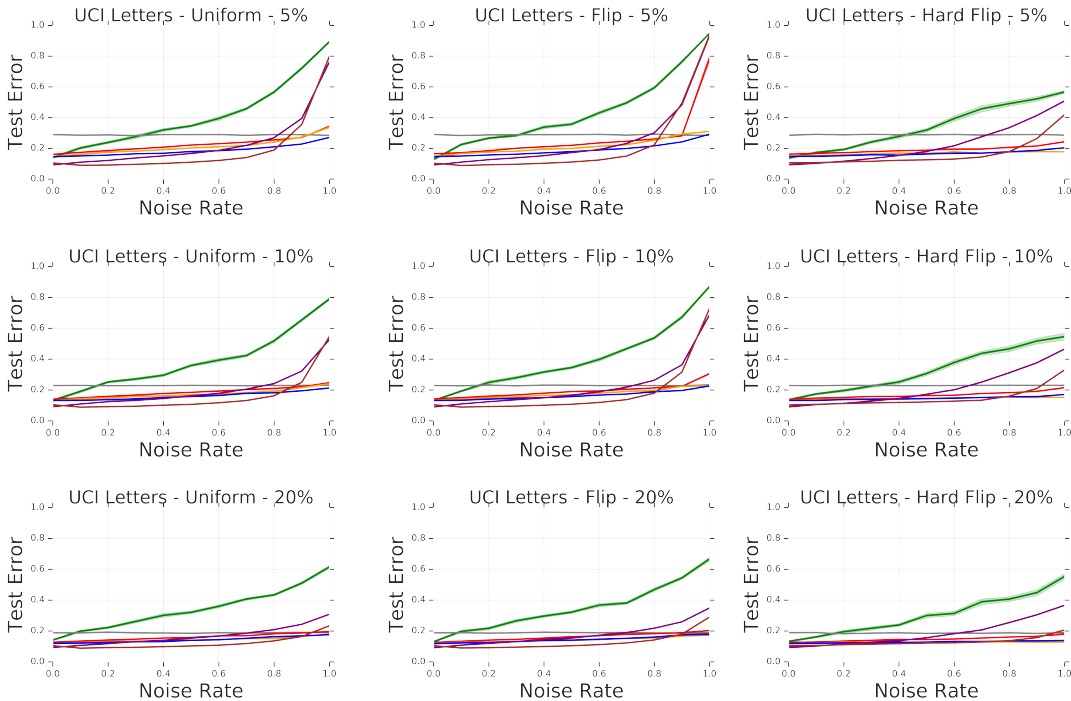

Figure 8: Plots for UCI Letters dataset at 5, 10, 20% clean data and all corruption types.

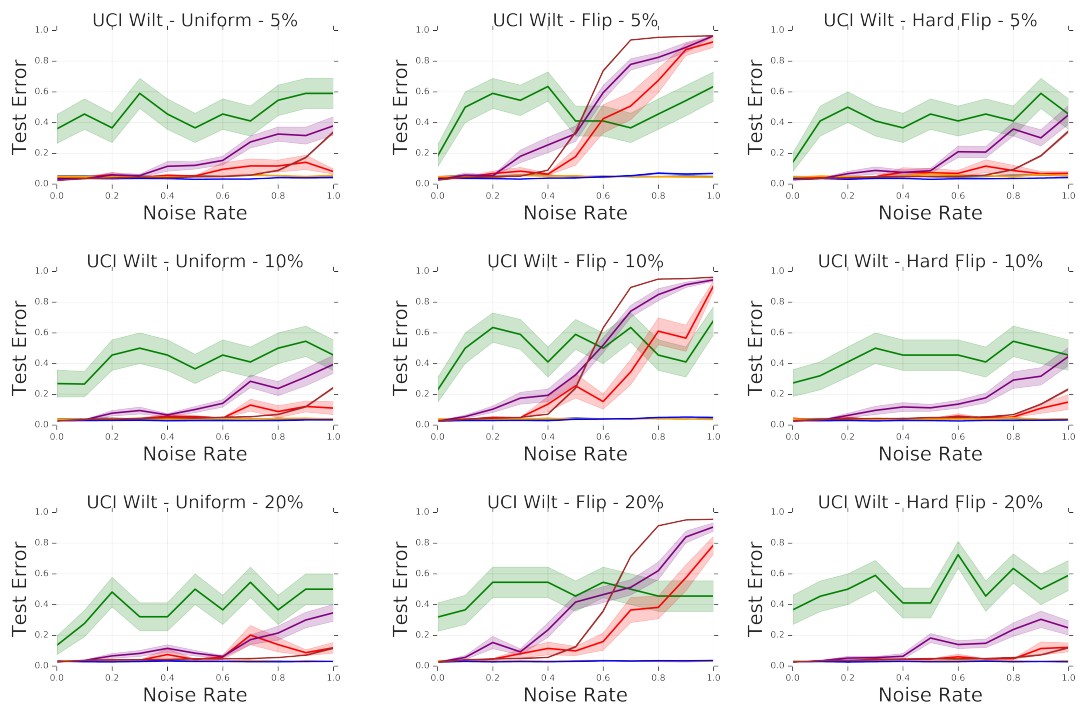

Figure 9: Plots for UCI Wilt dataset at 5, 10, 20% clean data and all corruption types.

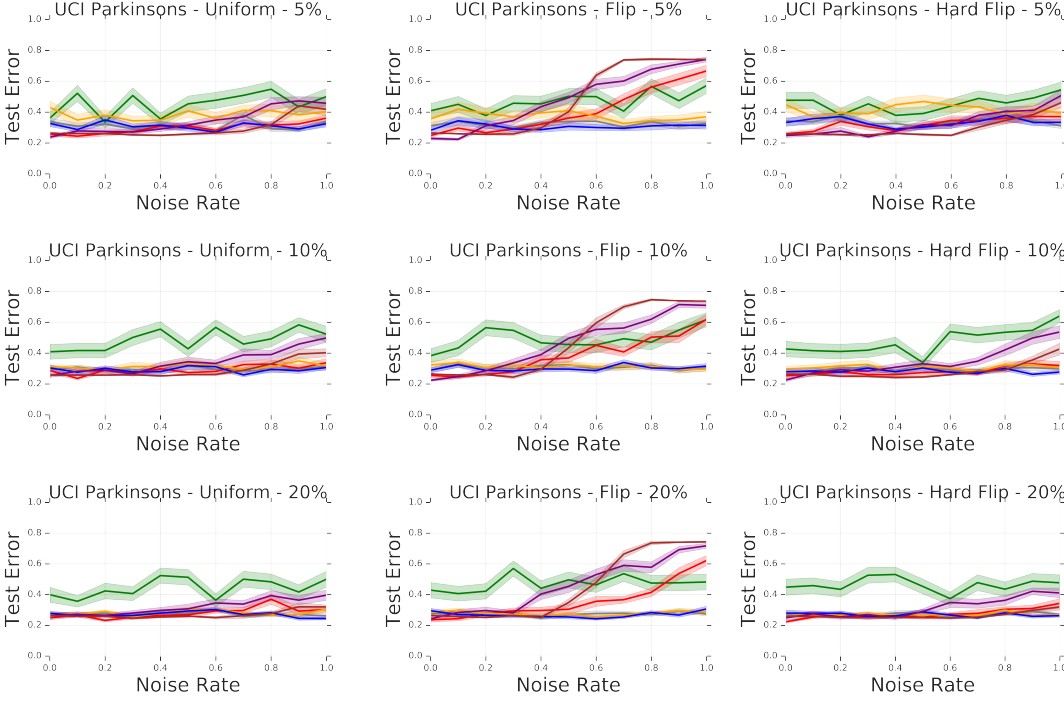

Figure 10: Plots for UCI Parkinsons dataset at 5, 10, 20% clean data and all corruption types.

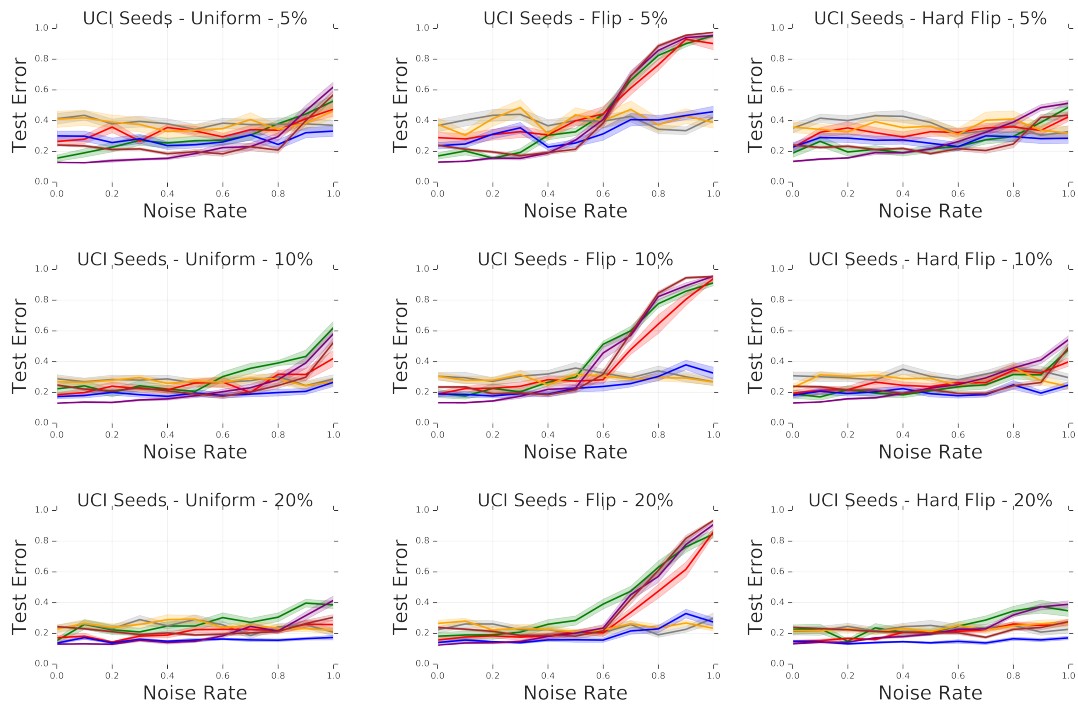

Figure 11: Plots for UCI Seeds dataset at 5, 10, 20% clean data and all corruption types.

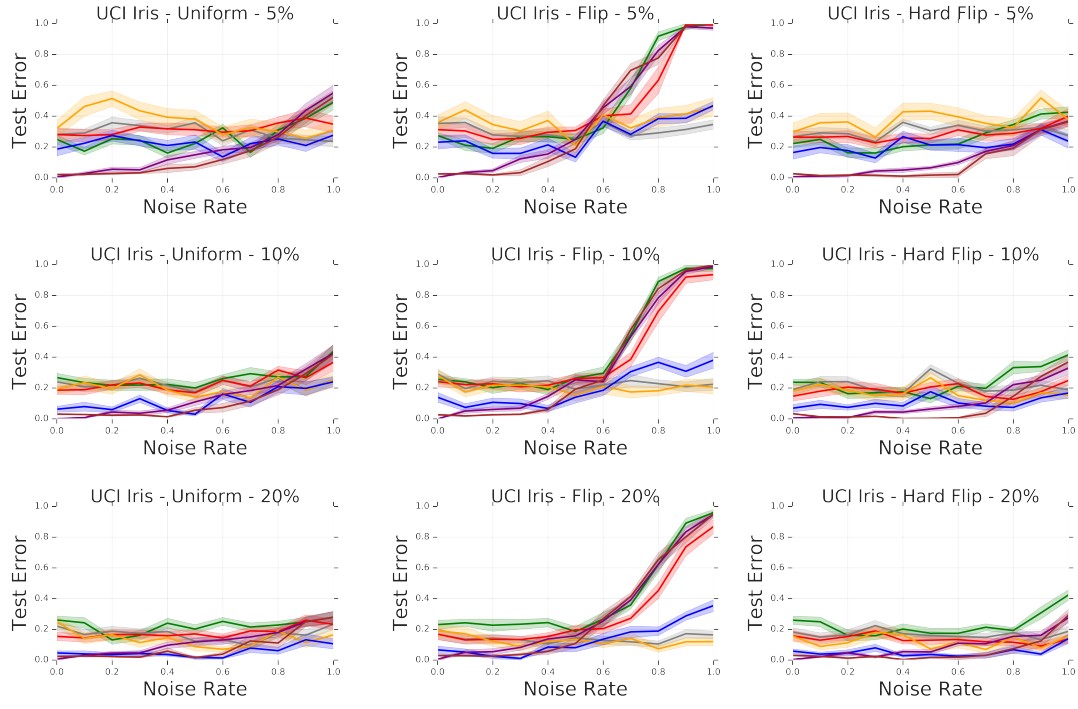

Figure 12: Plots for UCI Iris dataset at 5, 10, 20% clean data and all corruption types.

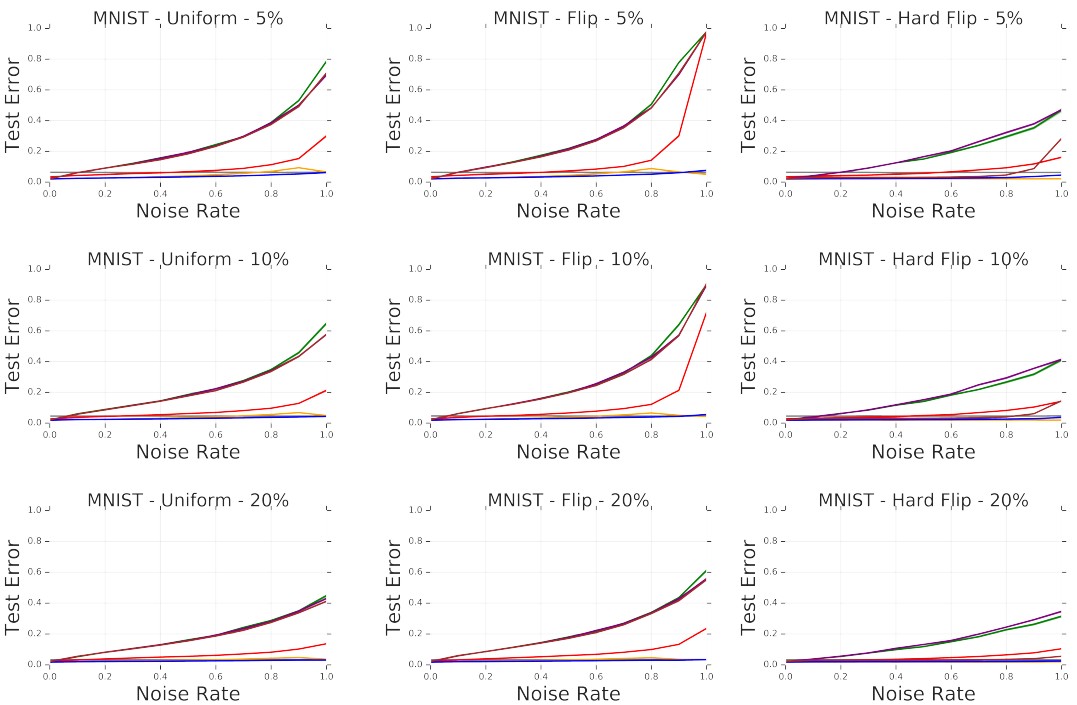

Figure 13: Plots for MNIST at 5, 10, 20% clean data and all corruption types.

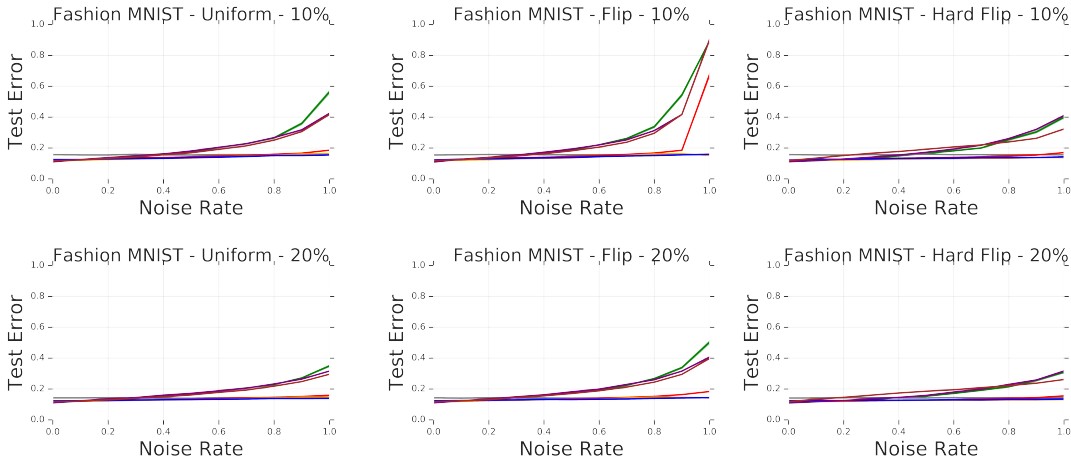

Figure 14: Plots for Fashion MNIST at 10, 20% clean data and all corruption types.

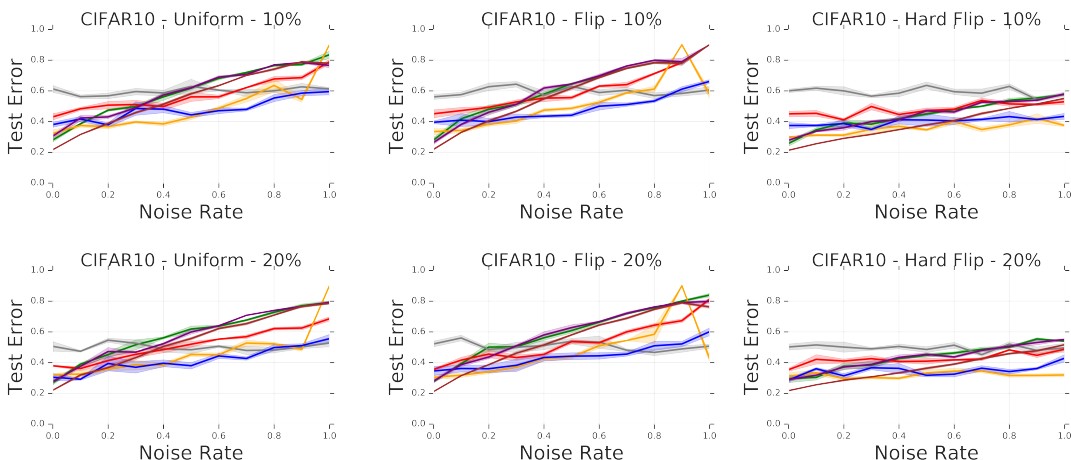

Figure 15: Plots for CIFAR10 at 10, 20% clean data and all corruption types.

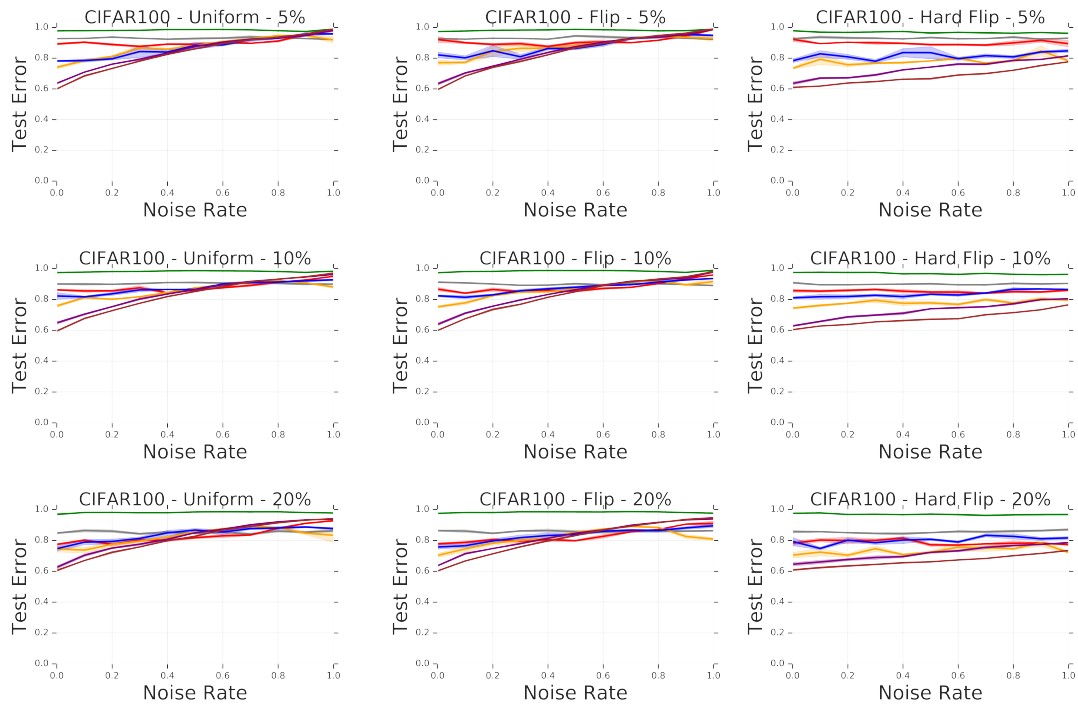

Figure 16: Plots for CIFAR100 at 5, 10, 20% clean data and all corruption types.

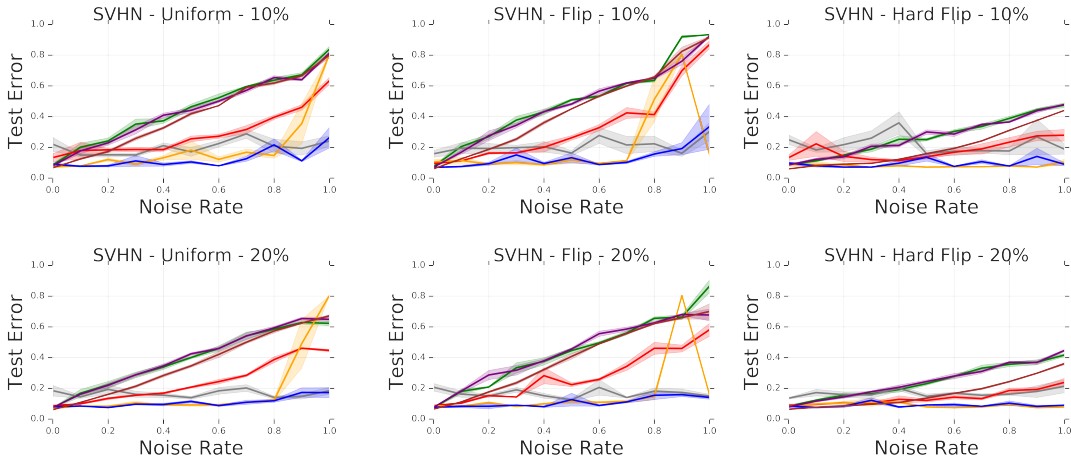

Figure 17: Plots for SVHN at 10, 20% clean data and all corruption types.

