# OpenReview forum: "Deep k-NN for Noisy Labels"
_ICLR.cc/2020/Conference — Reject_

### Official Review · AnonReviewer3 · 2019-10-16
**Official Blind Review #3**

**Rating:** 1

**Review:**

This paper provided "an empirical study showing that a simple k-nearest neighbor-based filtering approach on the logit layer of a preliminary model can remove mislabeled training data and produce more accurate models than some recently proposed methods". Even though it has many theoretical analysis and experiments, the paper itself is poorly written. There is no intuitive discussion on what is missing in existing methods, why the proposed method can be better, and when the proposed method may also fail.

Note that an important related work is missing, namely "Robust Inference via Generative Classifiers for Handling Noisy Labels" from ICML 2019 (see https://arxiv.org/abs/1901.11300). The idea of that paper is also making use of the learned representations of ANY discriminative neural classifier, where the geometric information of the hidden feature spaces can help to distinguish correctly and incorrectly labeled training data. That paper was a 20-min long oral presentation at Hall A (i.e., one of the most crowded sessions), and the authors should really compare with it both conceptually and experimentally.

**Experience Assessment:**

I have published in this field for several years.

**Review Assessment: Checking Correctness Of Derivations And Theory:**

I did not assess the derivations or theory.

**Review Assessment: Checking Correctness Of Experiments:**

I assessed the sensibility of the experiments.

**Review Assessment: Thoroughness In Paper Reading:**

I read the paper thoroughly.

---

### Official Review · AnonReviewer2 · 2019-10-24
**Official Blind Review #2**

**Rating:** 1

**Review:**

The authors propose to apply k-NN on the intermediate representations of neural networks for data cleaning. They prove some theoretical properties of k-NN and demonstrate that the proposed data cleaning approach is effective for some tasks.

It is recommended to reject the paper, with the following concerns in mind.

(1) The proposed approach is not deeply studied. For instance, what's the difference of applying k-NN on raw features, the earlier representations, the later representations, or even "all" representations? What's the effect of similarity/distance functions on the k-NN? Without the deeper study, Section 3 is at best a naive use of k-NN for data cleaning, and it is not clear whether the contribution is substantial.

(2) The theoretical analysis does not seem related to applying k-NN to *deep learning* intermediate features. It seems more related to applying k-NN in general. If so, it is also not clear how the theoretical analysis advances current knowledge about k-NN. Are the results original or known? What are the best theoretical results in the literature to compare with?

I thank the authors for answering about the originality. I agree that the original theoretical results is an important contribution on its own, but putting it in the context of deep learning is arguably not the best angle to present the contribution.

(3) It is not clear whether the experiments are compared with respect to state-of-the-art (or at least it is hard to see from Section 2). It seems that rather straightforward baselines are being compared.

I thank the authors for clarifying this.


**Experience Assessment:**

I do not know much about this area.

**Review Assessment: Checking Correctness Of Derivations And Theory:**

I assessed the sensibility of the derivations and theory.

**Review Assessment: Checking Correctness Of Experiments:**

I assessed the sensibility of the experiments.

**Review Assessment: Thoroughness In Paper Reading:**

I read the paper at least twice and used my best judgement in assessing the paper.

---

> ### Author Response · Authors · 2019-11-15
> **response**
>
> Re (2):
> The reviewer is correct that the analysis does not look at the intermediate features. Still, we believe that our representation-agnostic results are quite general and significant. They are general because the results say that given any initial representation, under mild assumptions, the kNN will be able to recover the noisy labels at least as well as any method up to logarithmic factors. The results are original --they use some insights from a recent analysis of kNN in the noiseless setting (Jiang 2019) and we provide a novel nonparametric assumption (kNN spread, Def 2) and show precise theoretical guarantees under this quantity as well as the other quantities in more classical results. Moreover, compared to previous works which analyze the noisy kNN setting (which we’ve cited), to the best of our knowledge, our results are the only finite-sample results while the previous works are asymptotic.
>
> Re (3):
> We consider Gold Loss Correction a state-of-the-art method and the other methods very competitive. We have added two new baselines as well.

---

### Official Review · AnonReviewer1 · 2019-10-24
**Official Blind Review #1**

**Rating:** 6

**Review:**

This paper proposes a k-NN method for identifying corrupted labels, and then applies this k-NN in the representation space of a deep neural net rather than the original feature space. Overall the paper is well written and the results look quite convincing

The theory appears to be important (if somewhat straightforward-looking) contributions of existing k-NN theory to the corrupted labels setting, based on the key quantity the authors defined as S_k, the minimum k-NN spread.

Since the theory highly depends on this quantity, the authors should after Definition 2 justify why they chose to base their results around S_k, and why intuitively it is the right quantity.

- I encourage another experiment where the label corruption is not completely at random, that is it depends on the values of x itself.

- In particular I encourage a synthetic experiment where the authors look at corruptions with varying S_k (minimum k-NN spread) values, to empirically verify the their theory holds and this really is a meaningful quantity to consider in the label-corruption setting.

- Why don't the authors show the results for vanilla deep kNN trained on the full (noisy + clean) dataset in their experiments. This seems important to ascertain the benefits that might be attributed to simply switching to kNN.  Or is deep kNN generally worse that the original model trained on the full dataset?

- Why didn't the authors show the original model trained on the full dataset?
Is it because it always does worse than all the baselines considered in the paper?
I would expect it sometimes does much better than Control (eg. when noise rates are low), and this is the straightforward approach must practitioners would use.

- Why don't the authors present the accuracy of the k-NN method at identifying corrupted datapoints vs the other methods that aim to explicitly identify the corrupted datapoints?
In general, it seems the authors did not compare other filtering baselines, which would be more related to their method, for example:

Learning with Confident Examples: Rank Pruning for Robust Classification with Noisy Labels
Northcutt et al. (2017). https://arxiv.org/abs/1705.01936


- It would be valuable to the scientific community if the authors can comment on:
Rolnick et al. (2018). Deep Learning is Robust to Massive Label Noise. https://arxiv.org/pdf/1705.10694.pdf

- Some similar looking ideas have been proposed in:

Gao et al. (2018). On the Resistance of Nearest Neighbor To Random Noisy Labels
https://pdfs.semanticscholar.org/4227/918020c15b719c415e93eb63d436583f1745.pdf

Parvin et al. (2010). A Modification on K-Nearest Neighbor Classifier.
https://globaljournals.org/GJCST_Volume10/7-A-Modification-on-K-Nearest-Neighbor-Classifier.pdf

so the authors should contrast their method/analysis against those papers.


- In Thm 1: "w.r.t. X" should be "w.r.t. x" (lower case)


**Experience Assessment:**

I have published one or two papers in this area.

**Review Assessment: Checking Correctness Of Derivations And Theory:**

I assessed the sensibility of the derivations and theory.

**Review Assessment: Checking Correctness Of Experiments:**

I carefully checked the experiments.

**Review Assessment: Thoroughness In Paper Reading:**

I read the paper at least twice and used my best judgement in assessing the paper.

---

> ### Author Response · Authors · 2019-11-15
> **Response**
>
> Thank you for your detailed review.
>
> Re: "Since the theory highly depends on this quantity, the authors should after Definition 2 justify why they chose to base their results around S_k, and why intuitively it is the right quantity."
>
> S_k is a natural way to quantify how spread out a set of points is in the k-NN setting which helps us quantify the impact of points with corrupted labels on the k-NN classifier. In the theoretical results, note that we only use S_2 to keep the statements simple. That is, S_2 is the minimum pairwise distance --we will clarify this in the text and stick with this simpler and perhaps more intuitive definition.
> -----------
> Re: "I encourage another experiment where the label corruption is not completely at random, that is it depends on the values of x itself. "
>
> We decided to focus on label-dependent noise, which we capture in our "hard-flip" corruption, since it's far more common than input feature-dependent noise both in practice and in the existing literature.
> -----------
> Re: "In particular I encourage a synthetic experiment where the authors look at corruptions with varying S_k (minimum k-NN spread) values, to empirically verify the their theory holds and this really is a meaningful quantity to consider in the label-corruption setting."
>
> This is a good suggestion. We have added a new section to the main text to study this on a two-Gaussian binary classification simulation where we insert corruptions on a grid. We show that as the grid width gets smaller, the more clean samples we need.
> -----------
> Re: "Why don't the authors show the results for vanilla deep kNN trained on the full (noisy + clean) dataset in their experiments. This seems important to ascertain the benefits that might be attributed to simply switching to kNN.  Or is deep kNN generally worse that the original model trained on the full dataset?"
>
> We have added this as a new baseline ("kNN-Classify") for all our experiments. We found that it performs better than the original model trained on the full data, but that it is still worse than our kNN filtering method, all in all.
> -----------
> Re: "Why didn't the authors show the original model trained on the full dataset?
> Is it because it always does worse than all the baselines considered in the paper?
> I would expect it sometimes does much better than Control (eg. when noise rates are low), and this is the straightforward approach must practitioners would use."
>
> This is a good suggestion. We have added this as a new baseline ("Full") for all our experiments.
> -----------
> Re: Why don't the authors present the accuracy of the k-NN method at identifying corrupted datapoints vs the other methods that aim to explicitly identify the corrupted datapoints?
> In general, it seems the authors did not compare other filtering baselines, which would be more related to their method, for example:
> Learning with Confident Examples: Rank Pruning for Robust Classification with Noisy Labels
> Northcutt et al. (2017). https://arxiv.org/abs/1705.01936
>
> This work, while relevant, focuses solely on binary classification while our method is applicable to general multiclass classification.
> -----------
> Re: "It would be valuable to the scientific community if the authors can comment on:
> Rolnick et al. (2018). Deep Learning is Robust to Massive Label Noise. https://arxiv.org/pdf/1705.10694.pdf
>
> - Some similar looking ideas have been proposed in:
>
> Gao et al. (2018). On the Resistance of Nearest Neighbor To Random Noisy Labels
> https://pdfs.semanticscholar.org/4227/918020c15b719c415e93eb63d436583f1745.pdf
>
> Parvin et al. (2010). A Modification on K-Nearest Neighbor Classifier.
> https://globaljournals.org/GJCST_Volume10/7-A-Modification-on-K-Nearest-Neighbor-Classifier.pdf
>
> so the authors should contrast their method/analysis against those papers."
>
> We have now commented on these papers in the Related Works.

---

### Decision · Program_Chairs · 2019-12-19

**Decision:**

Reject

**Comment:**

The paper proposed and analyze a k-NN method for identifying corrupted labels for training deep neural networks.

Although a reviewer pointed out that the noisy k-NN contribution is interesting, I think the paper can be much improved further due to the followings:

(a) Lack of state-of-the-art baselines to compare.
(b) Lack of important recent related work, i.e., "Robust Inference via Generative Classifiers for Handling Noisy Labels" from ICML 2019 (see https://arxiv.org/abs/1901.11300). The paper also runs a clustering-like algorithm for handling noisy labels, and the authors should compare and discuss why the proposed method is superior.
(c) Poor write-up, e.g., address what is missing in existing methods from many different perspectives as this is a quite well-studied popular problem.

Hence, I recommend rejection.